# The Impact of the SARS-CoV-2 Pandemic on the Needs of Non-Infected Patients and Their Families in Palliative Care—Interviews with Those Concerned

**DOI:** 10.3390/jcm11133863

**Published:** 2022-07-03

**Authors:** Christina Gerlach, Anneke Ullrich, Natalie Berges, Claudia Bausewein, Karin Oechsle, Farina Hodiamont

**Affiliations:** 1Palliative Care Unit, Department of Oncology, Hematology and BMT, University Medical Center Hamburg-Eppendorf, 20246 Hamburg, Germany; a.ullrich@uke.de (A.U.); kaoechsl@uke.de (K.O.); 2Department of Palliative Care, Heidelberg University Hospital, 69120 Heidelberg, Germany; 3Department of Palliative Medicine, LMU University Hospital, 81377 Munich, Germany; natalie.berges@med.uni-muenchen.de (N.B.); claudia.bausewein@med.uni-muenchen.de (C.B.); farina.hodiamont@med.uni-muenchen.de (F.H.)

**Keywords:** SARS-CoV-2, palliative care, family caregivers, pandemic, qualitative research

## Abstract

During humanitarian crises, such as a pandemic, healthcare systems worldwide face unknown challenges. This study aimed to explore and describe the effect of the SARS-CoV-2 pandemic on the needs of non-infected patients and family caregivers in specialist palliative care, using qualitative, semi-structured interviews. Data were analyzed using inductive content analysis, following the framework approach. Thirty-one interviews were conducted with patients/family caregivers (15/16) in palliative care units/specialist palliative home care (21/10) from June 2020 to January 2021. Well-known needs of patients and family caregivers at the end of life remained during the pandemic. Pandemic- dependent themes were (1) implications of the risk of contagion, (2) impact of the restriction of social interactions, (3) effects on the delivery of healthcare, and (4) changes in the relative’s role as family caregiver. Restriction on visits limited family caregivers’ ability to be present in palliative care units. In specialist palliative home care, family caregivers were concerned about the balance between preserving social contacts at the end of life and preventing infection. Specialist palliative care during a pandemic needs to meet both the well-known needs at the end of life and additional needs in the pandemic context. In particular, attention should be given to the needs and burden of family caregivers, which became more multifaceted with regards to the pandemic.

## 1. Introduction

In early 2020, SARS-CoV-2, a viral disease first described in 2019, became epidemic in China and spread across borders, being classified as a pandemic by the WHO in March 2020 [1]. SARS-CoV-2 poses the highest risk to older adults and people with serious illnesses [2,3,4] - central characteristics of the vulnerable palliative care population. Little is known about the needs and fears of seriously ill and dying patients, or their family caregivers (i.e., family and friends), in humanitarian crises such as a pandemic, especially in the context of specialist palliative care. This applies to patients dying from their underlying disease, as well as those dying from SARS-CoV-2. To ensure the highest possible quality of life, even in the current crisis, it is necessary to understand how the pandemic affects the needs of those concerned.

When considering what preoccupies patients at the end of life and what their needs are, the fundamental complexity of this situation becomes apparent. Physical symptom control represents only the first obvious aspect that may be affected by the SARS-CoV-2 pandemic. The structures, processes, and workforce capacities of palliative care are strained. Although specialist palliative care services responded rapidly to the SARS-CoV-2 pandemic [5], staff shortages and struggles to obtain essential equipment and medicines occurred frequently [6]. Consequently, palliative care units and hospices were even closed, despite constant demand for inpatient palliative care [7,8,9].

Compared to pre-pandemic times, general practitioners indicated a decline in quality of end of life care in almost 40% of their patients [10]. However, the accessibility, quality, and continuity of the palliative care setting are essential for severely ill and dying patients [11]. Furthermore, preservation of social stability and normality are known to affect the quality of life of these patients [11], and patients rely heavily on family caregivers for support, care, and social contacts [12]. During the SARS-CoV-2 pandemic, however, patients were predisposed to social isolation, due to physical distancing policies and restricted in-person visitations, especially when hospitalized.

Beyond its effects on patients, the SARS-CoV-2 pandemic had a profound impact on family caregivers of severely ill and dying patients. Informal social, emotional, and practical support for family caregivers has been constrained by social distancing policies, restricted travel, quarantine, and self-isolation. The reluctance of family caregivers to ask for help might even have increased, given the fears of having people coming to their home who could spread the virus [13]. Consequently, many family caregivers continue to deliver complex care unsupported when a family member is at the end of life [14]. Considering family caregivers’ experiences of disruptions to their social relationships and loneliness in non-pandemic times [15], it is likely that SARS-CoV-2 may exacerbate their risk of social isolation. Furthermore, family caregivers were no longer allowed to meet their loved ones in hospitals or healthcare facilities. Disruption to the communication between patients at the end of life and their family caregivers can have significant consequences, including family members developing depression, anxiety, and complicated grief reactions [16,17].

We aimed to explore and describe how the experience of patients and their families not infected with SARS-CoV-2 receiving specialist palliative care has been affected by the exceptional pandemic situation.

## 2. Materials and Methods

### 2.1. Study Design and Context

We conducted a qualitative study interviewing patients without infection and receiving specialist palliative care in hospital wards and at home, as well as their family caregivers, in Hamburg and Munich, representing the North and the South of Germany.

This qualitative study was part of a national mixed-methods project, aiming at developing a National Strategy for Palliative Care of Severely Ill and Dying People and their Relatives in Pandemics (PallPan) in Germany [18]. This research is based on critical realism, where reality, as an independently existing dimension, is accessible by observation of individual perceptions and contextual interpretations [19]. We used the consolidated criteria for reporting qualitative studies (COREQ) framework to report on the design, analysis, and results of our study [20].

### 2.2. Participants and Settings

We followed a purposive sampling strategy, including adult patients and family caregivers in specialist palliative care in a palliative care unit or at home. The sampling framework (see Appendix A) allowed for a heterogeneous sample, ensuring maximal variation of information [21].

Persons not fluent in German and those considered to be too burdened, as judged by their clinical team, were excluded. Participants were recruited in two German university hospitals. Eligible participants were approached by the clinical teams. Those interested in participation were contacted by the researchers and interviewed in person or via telephone, based on interviewee preference or contact regulations. All participants provided written informed consent.

### 2.3. Data Collection

The interview guide included open-ended questions related to the participants’ perceptions of their needs and experiences of care during the pandemic (Table 1 and Appendix A).

The interview guide [18] was developed following the four-step approach by Helfferich [22]. Interviews were conducted by two researchers (NB, AU), neither of whom had prior relations with the participants. Questions were phrased sensitively, in accordance with techniques for interviewing vulnerable people in palliative care [23,24]. Interruptions were possible at any time during the interview, at the request of the participant. Field notes were taken during each interview.

### 2.4. Data Analysis

We applied content analysis supported by the “framework approach” [25,26] to analyze descriptive accounts and summarize phenomena across individuals. Audio-recordings of interviews were transcribed verbatim and managed using the software package MaxQDA18. Data were analyzed using qualitative content analysis, applying the framework approach [26] and subdividing the analysis into three steps: (1) data management, (2) description, and (3) explanation. The framework was developed inductively. The first analysis step provided an overview of the data content and saturation. The initial framework sorting the data was provided by the interview guide structure. In the second step, the spectrum of information was mapped, categories developed, and key dimensions identified by FH and NB, based on five interviews. After completion of the data collection and determining all categories, the third step was to develop explanations. This was an iterative process carried out by NB, AU, CG, and FH. A coding guide and the verification of intra-rater reliability (NB) for three interviews, and inter-rater reliability (NB, AU) for five interviews, ensured coding consistency. Inconsistencies between coders were reconsidered, discussed in two research group meetings, and alternative interpretations were incorporated into the analysis.

### 2.5. Ethical Approval, Registrations

The study was approved by the Research Ethics Committee at the Ludwig-Maximilians-University Munich (Ref: 20-403) and by the Research Ethics Committee of the Medical Council of Hamburg (Ref: 2021-746582-BO-bet), Germany. This study was registered on the German Clinical Trials Register (DRKS00023839).

## 3. Results

### 3.1. Demographic and Interview Characteristics

We conducted 31 interviews with 15 patients and 16 family caregivers between June 2020 and January 2021. Five people declined interviews due to psychological burden (three patients) or without giving a reason (one patient, one family caregiver). The characteristics of the 31 participants are presented in Table 2.

Interview duration ranged between 9 and 75 min (Md 30.0, IQR 22) for patients and between 10 and 87 min (Md 36.5, IQR 32) for family caregivers. No interview was stopped on demand of the participant. Twenty-four interviews were conducted face-to-face and seven by telephone.

### 3.2. Living in Times of SARS-CoV-2

Strikingly, patients often denied being affected by the virus when asked directly, but implicitly expressed great concern in the course of the interview.

*“Corona has no meaning for me. What is important for me now is simply that, yes, at some point I will close my eyes forever because of this disease [COPD]. And that there… there is nothing stopping me.”* (4th minute of the interview) “*(…) But right at the beginning [emergency room] it was cruel that my wife, for example, was bringing a bag with stuff, she stood in front of the door and was not allowed in. That is inhumane (…)*” (16th minute of the interview) (Patient 2_028; palliative care unit)

Furthermore, many patients and family caregivers voiced concerns about the global and personal impact of the pandemic and, eventually, its meaning for humanity.


*“So what I notice is that through COVID (…) there is a latent fear of health hazards [present] that I never had explicitly like this.”*
(Family caregiver 2_030; palliative care unit)


*“Why is this spreading all over the world? From north to south, from east to west. It makes you think, right. (…) Well, now it’s here and we have to live with it.”*
(Patient 2_027; palliative care unit)

These insights relate to human experiences during a global crisis, narrated by people who—beyond their “being a severely ill and dying patient” or “being a family caregiver”—are human beings faced with unprecedented challenges.

### 3.3. The Experience of End of Life Remains Unique

We identified two types of needs, those independent of the pandemic and those associated with the pandemic. Although the interview focused on the palliative care experience in the pandemic, patients and family caregivers addressed needs that are paramount in the palliative care situation and independent from external conditions. Being at the end of life was an exceptional experience, shaped by well-known dimensions, which we called pandemic-independent needs. Themes and subthemes are listed in Table 3 (for example quotes see Appendix A).

While basic palliative care needs did not cease to exist, patients and family caregivers named new needs associated with the pandemic (Figure 1). Themes related to the pandemic were (1) implications of the risk of contagion, (2) impact of the restriction of social interactions, (3) effects on the delivery of healthcare, and (4) changes in the relative’s role as family caregiver.

#### 3.3.1. Implications of the Risk of Contagion

Patients and family caregivers described a variety of ways in which possible SARS-CoV-2 infection affected them. Most patients reported not to care about being infected by SARS-CoV-2, some felt ambiguous. However, some patients perceived the virus as a second enemy and protested against the disproportionate danger of becoming a victim of the pandemic whilst fighting cancer.


*“Personally, I’m not so much afraid of COVID-19 right now… the present story may also weigh a little more, because it is more tangible, while COVID is still not really that tangible. I mean, it is THERE, but I simply don’t happen to have it. I do have lung cancer though.”*
(Patient 2_006; palliative care unit)


*“And I’m always afraid for my [NAME SON], when he goes shopping or something, that he won’t come home healthy and so on. What do I DO then? These are fears, you know?”*
(Patient 2_019; specialist palliative home care)

In contrast, many family caregivers worried about themselves, visitors, or healthcare professionals being a risk for the vulnerable patients.


*“The thing with COVID is, of course, that if I got infected, I would miss out here as well. Meaning that he would need (…) a professional nurse, who cares for him. Even worse, well yes, if he catches the virus, what happens then? Then he might die of the virus because I was contagious?”*
(Family caregiver 2_021, specialist palliative home care)

However, no patient reported worrying about being contagious themselves, and actually being a risk for their family caregivers while spending quality time. The patients found their own illness and fate more relevant than the pandemic.

#### 3.3.2. Impact of the Restriction of Social Interactions

Patients focused on the challenge they saw in their changed role in their own and others’ life, enhanced through social distancing. The simple reality that the risk of contagion is associated with social interaction shaped the patients’ experience. Experiencing sudden contact restrictions and no-visitor-policies made them feel lonely:


*“Suddenly I was so lonely, all the time. EVERYONE had withdrawn. (…) There was no phone call anymore. As if, everything was INFECTED somehow. (…) I can’t even express what it felt like. The loneliness and being alone, and that no one is there anymore.”*
(Patient 2_019, specialist palliative home care)

Many patients were concerned with being able to see their loved ones safely and stated they had not met many of their family or friends since the onset of the pandemic. This was especially true for hospitalized patients, who described severe isolation because of quarantine policies and strict visitor-restrictions.

Family caregivers linked social isolation to observed declines in patients’ mental and physical health. Sadness and depressive symptoms were the most mentioned problems:


*“….and that [visitor restrictions] was of course very, very hard for her. This made her mentally exhausted. She constantly said how much she suffered from it. I totally understood. Stunningly, her condition really did get worse from the time the pandemic broke out.”*
(Family caregiver 2_011, palliative care unit)

Patients and family caregivers reported heavily burdening and traumatizing situations due to tight regulations for visitors, leading to acute mutual worries about the other’s wellbeing, loss of sense of control, anger, and fear, resulting in increased and lasting emotional stress.

*“At the very beginning [in the emergency department], it was cruel that, for example, my wife brought me a bag with some things, she stood in front of the door and was not allowed in. That’s inhuman. So, whoever decided that has no brain, he has never thought about what that means for families.”*.(Patient 2_028, palliative care unit)

In this context, family caregivers reported the negative impact on their confidence in the patient’s best care:


*“…if the person (…) had said ‘you know what don’t you worry, I won’t leave your husband alone before I know that the next one will take care of him’ or whatever. Instead, I felt, I let him go, the door [of the emergency room] opens, the door closes and he is gone (…) I’ve become a little less trusting. It’s more difficult for me to say, everything will be fine, and they will take care and they will do what they can.”*
(Family caregiver 2_030; palliative care unit)

However, one family caregiver also indicated a positive aspect of the stay-home-order, allowing her to spend more time with her loved one than usual:


*“Without COVID-19 I am not sure whether I would have had time to accompany my mother in her last phase to the extent that I am currently able to do.”*
(Family caregiver 2_011, palliative care unit)

Patients and family caregivers claimed limitations of telecommunications when compared to in-person contact.


*“I do have a social sphere and a family too, and they also call every day and ask how things are going. (…) Those are the contacts outside. That helps me to bear it all. (…) I mean, today you can also look at each other via WhatsApp and so on, but it’s just different if you can only see each other or touch each other.”*
(Patient 2_029, palliative care unit)

#### 3.3.3. Effects on the Delivery of Healthcare

Patients and family caregivers described a variety of ways in which the pandemic situation disrupted access to healthcare services, particularly in non-specialist palliative care. Mobile patients receiving palliative and supportive care in outpatient clinics reported worries or actually disrupted care, which worried them in terms of their prognosis and reliable care.


*“For example, I wanted to see my doctor. And called him, that’s a pulmonologist. And they said the practice would be closed for 14 days due to Corona, bang, out. Who will take you? Where else can you go? Which doctor is willing to listen to you or take you in? THAT worried me a bit.”*
(Patient 2_006, palliative care unit)

However, one family caregiver perceived that integration of specialist palliative home care was delayed.


*“I think we already asked once in March, my husband, about this palliative home care situation. And we were told ’Yes, it’s really bad at the moment, unfortunately we can’t do it. Because of Corona.’ (…) He was still undergoing radiation therapy, of course. That was a bit of a step backwards, because we were hanging in the air a bit, because he was already in pain. So if Corona hadn’t been there, maybe they could have supported us earlier?”*
(Family caregiver 2_021, specialist palliative home care)

Patients chose to avoid care in hospital settings because they feared that visitor restrictions would prevent them from seeing their family caregiver. Some voiced that they decided to postpone or cancel treatments, to avoid a setting in which family caregivers would not be able to accompany them:


*“That is why I’m so happy that I ended up here on the palliative care unit and not on an oncology ward. Because there was also the question of whether I would have chemotherapy, the last cycle, and I thought, no (…). I’m staying here and then I won’t have it done, that’s just the way it is.”*
(Patient 2_029, palliative care unit)

However, many patients voiced positive experiences regarding the quality of specialist palliative care, both inpatient and at home. When inpatient care became necessary, the palliative care units represented a refuge. Despite the restrictions, no one felt abandoned, and the feeling of being well looked after remained.


*“And as I said, I have ALWAYS had wonderful care. I didn’t suffer more under it than I did before in my illness. I say, it was a surreal dream. But now, I never had the feeling that I was or had been DESERTED.”*
(Patient 2_016, specialist palliative home care)


*“But this one-to-one care was still quite normal for her. (…) There were always regular calls from Mrs. (NAME) or the doctor (NAME), whether everything was still working. And if something happened, someone still came to the place and looked after her straight away, (…) So everything was still as usual, the care was simply well-regulated to one hundred percent.”*
(Family caregiver 2_023, specialist palliative home care)

Most patients accepted protective equipment like clinical masks and other hygienic procedures and perceived them as inevitable, even in intimate care situations and when faced with sensitive information.


*“…with all the COVID-19 swab testing (…) we have to go through that, I am not afraid at all. I’ve already been tested like 8, 9, 10 times and I do hope it stays like this…that I continue to be negative [for SARS-CoV-2].”*
(Patient 2_029, palliative care unit)

Family caregivers also acknowledged the precautionary measures to avoid spreading the virus, but they demanded more and precise information about protective equipment, specific regulations, and procedures.


*“It would be easier if they were clearer. (…) Although it might be a little uncomfortable at first, (…) when someone says ‘Please understand that due to the pandemic, the rules are so and so for safety reasons (…)’ Friendly, factual, short and precise.”*
(Family caregiver 2_030, palliative care unit)

Additionally, family caregivers experienced an imbalance of power between themselves and those who decide over hygienic procedures, regarding both healthcare professionals and security staff. Family caregivers who are in a vulnerable situation felt exposed to non-supportive others, who were controlling them:


*“However, it happened to me that one of the nurses shot up to me and said in a rough attitude ‘Who are you and what are you doing here?’ I had already registered myself at the reception and looked like a fool, feeling like a school kid in a test. I experienced that moment as very unpleasant. And also, as very lecturing.”*
(Family caregiver 2_030, palliative care unit)

#### 3.3.4. Role of the Relative as Caregiver

Physical social distancing also had a psychological dimension, interfering with traditional attachments and the supportive network of family and friends, as well as professional services. When able to spend time with their loved ones, family caregivers were grateful for the opportunity to stay connected with the patient, but also reported burdening and upsetting aspects.


*“Well, it’s always like this, after all only I can come and that only here. Just me. And that is sometimes very stressful (…) But of course I would be so relieved, if someone just once said ‘And today you only have to do two hours, because the other three hours I’m going this time.’ And that it could have been shared a bit, that maybe a second person had access. That’s so—so, not just one person per day but sometimes maybe two, then you of course have to decide again who the second person is.”*
(Family caregiver 2_031, palliative care unit)

Being the person on whom the patient’s care crucially depended also increased the pressure of responsibility regarding the risk of infection. In addition to the risk that family caregivers could infect the patient, there was also the risk that family caregivers could become infected with SARS-CoV-2, which would result in failure to provide care. Holding the balance of contacts between distancing and accompanying became one of the biggest challenges for family caregivers. In palliative home care, family caregivers felt responsible for the best care of the terminally ill family member. However, the focus and definition of best care differed considerably between families. Some complied with the recommendations regarding hygienic measures and social distancing, while others voluntarily disregarded these, because they attached greater importance to loving contact with the patient than to the risk of lethal SARS-CoV-2 disease in the face of the life-limiting underlying disease.


*“My decision was really to weigh it up and say, okay, if my mom WOULD get infected and WOULD get pneumonia and die from it, then I MUST, can and want to take responsibility for it. Because otherwise she might not for weeks, maybe months see the persons who are extremely important to her and who contribute to life sustainment at a different level and who are, yes, essential for her quality of life. Finally, I very clearly accepted the risk of shortening life.”*
(Family caregiver 2_018, specialist palliative home care)

Family caregivers whose loved ones became patients in palliative care units or other inpatient facilities felt confronted themselves with visitor restrictions. Family caregivers reported this as a distressing period, since they could not fulfil their tasks as family caregivers, such as practical caregiving, monitoring the patient’s condition, supporting the patient’s social connectedness, or simply holding the patient’s hand. Most importantly, the perception of restrictions as resulting in being stopped from accompanying the terminally ill family member corresponded well with the patients feeling of being lonely.


*“With regard to COVID-19, I have to say that it really was difficult for me because I was unable to maintain any contact with her [the mother] at all. In the meantime, (…) her dementia progressed, thus she was in the protected department [of a nursing home], and so I could not reach her by phone. I haven’t been able to visit her since she was moved there.”*
(Family caregiver 2_002, palliative care unit)

## 4. Discussion

We explored the experience of non-infected patients and family caregivers facing a palliative care situation during the SARS-CoV-2 pandemic in palliative care units and specialist palliative home care in two German cities. The results of this study informed healthcare professionals, providers, and policy via the National Strategy for Palliative Care of Severely Ill and Dying People and their Relatives in Pandemics (PallPan) in Germany [18,27,28].

Palliative patients and family caregivers reported unmet needs regarding coping with the risk of contagion, social interaction being affected by social distancing, effects of the pandemic on healthcare delivery, and the challenged role of family caregivers increasing the burden of responsibilities. Nevertheless, the well-known needs of patients and their family caregivers at the end of life remained.

The qualitative approach allowed for an in-depth exploration of patients’ reactions, which extended beyond their explanations and captured what they were less aware or even unconscious of. From this, we found that, in daily clinical practice, a discussion process is important to capture the concerns and burdens of patients receiving palliative care during the pandemic.

### 4.1. Pandemic-Independent Palliative Care Needs

A systematic review synthesizing all patient-valued dimensions of palliative care found eight aspects relevant from the patients’ perspective: physical, personal autonomy, emotional, social, spiritual, cognitive, healthcare, and preparatory [11]. Although this was not the primary aim of the study, we retrieved these very aspects in all interviews in the palliative care situation during the pandemic. Thus, people knowingly approaching the end of life are encapsulated in a unique experience, where every individual goes through independent of external conditions, such as the pandemic. Our results are in line with numerous articles published in recent months, highlighting that general palliative care needs have not changed in the pandemic [29,30]. However, to the best of our knowledge we are the first reporting voices from patients receiving specialist palliative care in Germany.

### 4.2. Pandemic-Dependent Palliative Care Needs

While patient worries in pandemic and non-pandemic times were largely comparable, family caregivers differed in some aspects from patients regarding their worries during the pandemic. Worries and fears are repercussions of the end of life situations experienced by any family caregiver, but these are exacerbated by the risk of contagion with SARS-CoV-2 and the potential consequences for all affected. First quantitative studies with family caregivers reported the majority feared that the cancer patient may contract SARS-CoV-2 [31], or that they themselves may contract the virus, thus not being able to care for their chronically ill patient [32]. Furthermore, anxiety and guilt may be experienced, for reasons such as transmission of SARS-CoV-2 to the patient [13].

Patients and family caregivers reported isolation, feelings of loneliness, and traumatic situations, due to social distancing and visitor restrictions. Considering the significance of connectedness at the end of life [33], and the present additional stressors caused by the pandemic circumstances, the negative effects of constrained connections between patients and family caregivers must not be neglected. These include the increased burden and stress-related symptoms [34], decreased quality of communication with healthcare professionals [17], compromised dignity [35], and long-lasting grief [36]. For family caregivers, double-hit effects of the pandemic, due to the risk of contagion and risk of social isolation, have been reported [37].

Regarding the delivery of healthcare, family caregivers reported their need for clear and detailed information about infection prevention, rules, and procedures when a patient is cared for in an inpatient setting. Others found caregivers’ informational needs regarding hospital regulations and procedures to be very important, both in light of visitor restrictions due to the pandemic [38], as well as in pre-pandemic times [39]. Overall, patients and family caregivers reported poorer end of life care and more pandemic-related challenges in healthcare settings other than specialist palliative care. Therefore, our results reflect the importance of maintaining a person- and family-centered approach to palliative care, to conquer the effects of the pandemic, as patients and family caregivers felt supported enduring visiting restrictions, infection control measures, and altered care arrangements.

Our study showed that family caregivers continued to deliver complex care, but had to cope with unlimited responsibility, additional duties, and novel challenges in the support of the severely ill or dying patients. Family caregivers have an essential role in enabling the connectedness of patients. Family caregivers for other severely ill patients reported similar experiences, for example end-stage renal disease [40], Alzheimer’s dementia [41], neurodegenerative diseases [42], hematopoietic stem cell transplantation [43], and cancer [29]. However, the findings of this study highlighted the conflicts of family caregivers between minimizing the patient’s risk of contagion with restricted visits, and supporting connectedness at the end of life.

### 4.3. Clinical and Research Implications

The results clearly emphasize the need of personal contact between patient and family. Both, patients and family caregivers referred to the burden of isolation. Facets of these burdens included feelings of loneliness, mutual uncertainty, and worries about the loved one’s wellbeing. Furthermore, being the only visitor for a severely ill and dying patient can be a burdening experience. Accordingly, to support a feeling of certainty in an uncertain situation, a visiting concept for family caregivers of severely ill and dying people should be developed, granting regular visits by non-infected family caregivers jointly or alternatingly. Furthermore, regulations on how infected family caregivers can be enabled to visit are required. Of paramount importance is communication of regulations to patients and family caregivers. When personal visits by the family are not possible or are only possible to a limited extent, means of communication should be provided for establishing communication and closeness, e.g., video calls. The use of these means should be offered proactively and at a low threshold, including having team members ready and trained to support patients and family caregivers less familiar in using electronic communication devices [27,28]. Kuntz et al. implemented and evaluated electronic family meetings as part of specialist palliative care consultations. With their study, they showed, not only that the use of virtual means is feasible and acceptable for patients and their families receiving specialist palliative care, but also how these helped to integrate palliative medicine in the care of severely ill patients during the pandemic [44].

Our data indicate that family caregivers are worried about not being able to fulfil their duties. Family caregivers have a crucial role in caring for severely ill and dying people [45], and many patients, as well as family caregivers, report home as the preferred place of care [46]. Furthermore, the need to understand and support family caregiver needs, in order to enable them to take on care activities and support the specialist teams is recognized [47]. During the pandemic, caregiving is associated with additional challenges, because family caregivers are facing highly unusual circumstances and a disruption of their caregiving routine. We need research on how to best meet the patient’s and family caregiver’s needs, tailored to the diversity at local levels, and how, when, and where to allocate resources to palliative care.

The importance of supporting family caregivers becomes even more pressing when looking ahead to future societal challenges. The workforce of specialist palliative inpatient and outpatient teams was challenged in the pandemic, in addition to the demographic gap identified so far [48,49,50]. How the current projections, building on the increasing number of people with age-related diseases, such as chronic organ failure, dementia, and cancer can best integrate infectious diseases should be further investigated. Undoubtedly, the health care system will be facing even greater challenges, and the need for family caregiver support will grow.

### 4.4. Strength and Weaknesses of the Study

To date, the pandemic has seen a series of SARS-CoV-2 waves and interviews were conducted during the first wave; both during the most severe lockdown and the greatest restrictions, and at several stages of lifting the restrictions, as well as during the second wave, when restrictions were gradually reintroduced in Germany. The experience of patients and family caregivers may have been impacted by these dynamics. However, this study may best reflect the direct impact of the pandemic on patients and their family caregivers receiving specialist palliative care. Both sites are metropolitan, and the results may not fit for patients and family caregivers in non-metropolitan areas; however, the purposeful sampling according to the sample framework ensured high socio-demographic variation. Access to the study may have been limited, because there were no options for self-recruitment, partially due to pandemic restrictions. Patients and family caregivers in specialist palliative home care are particularly hard to reach. We were able to overcome these obstacles by approaching patients and family caregivers in the care of the services associated with the researchers’ institutions, who are experienced in patient assessment and accept research. The latter would also be a safeguard against the risk of gate keeping associated with the exclusion criterion of patients or family caregivers judged too burdened for interview participation. However, we did not systematically record all patients admitted to the services for exclusion criteria, because the purposive sampling focused on socio-demographic variation, rather than a high number to reach interview content saturation.

## 5. Conclusions

The exploration of the impact of the pandemic on the patient’s and family caregiver’s experience of specialist palliative care at home and in hospital resulted in two main outcomes. First, basic palliative care needs do not change during a pandemic. Second, patients and family caregivers in specialist palliative care have additional needs and burdens, depending on pandemic challenges. Thus, maintaining healthcare services, including specialist palliative care, during the pandemic needs not only to focus on the contagious and infected patients, but also on those who are already under care in existential situations. Some interventions may be postponed, some patients can wait, but at the end of life, impeccable care for patients and support for family caregivers cannot wait, because there is only one chance.

## Figures and Tables

**Figure 1 jcm-11-03863-f001:**
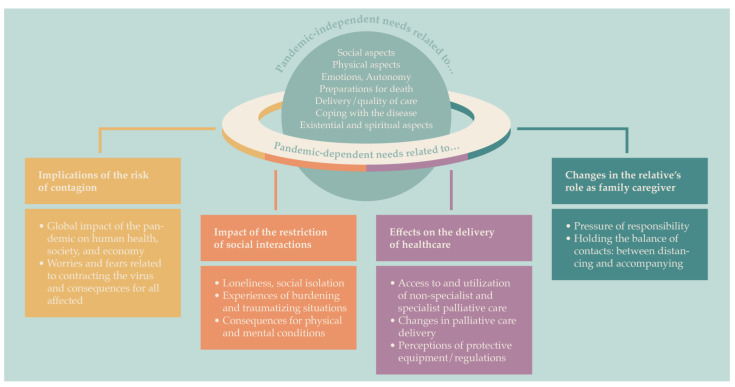
Themes of pandemic-dependent needs.

**Table 1 jcm-11-03863-t001:** Topics of the semi-structured interview guide.

Introducing the topic and conduct of the interview Exploration of current themes and needs of patients and family caregivers Perceptions of how the pandemic impacted the everyday life of patients and family caregivers Pandemic-related worries and fears of patients and family caregivers Consequences of the pandemic on the patient’s care Internal and external resources supporting patients and family caregivers in the current situation Closing of the interview

**Table 2 jcm-11-03863-t002:** Characteristics of patients and family caregivers.

		Palliative Care Unit		Specialist Palliative Home Care	
		Patients (*n* = 10)	Family Caregivers (*n* = 11)	Patients (*n* = 5)	Family Caregivers (*n* = 5)
Gender (*n*)	Woman	6	9	3	4
	Man	4	2	2	1
Age (M; range)	Age in years	69.3; 55–91	51.1; 31–77	72.0; 57–88	54.8; 46–67
Family Status (*n*)	Single	6	5	3	3
	In a relationship	4	6	2	2
Relationship (*n*)	Partner	─	4	─	2
	Daughter/son	─	3	─	3
	Others	─	4	─	─

**Table 3 jcm-11-03863-t003:** Themes and subthemes of pandemic-independent needs.

**Needs concerning physical aspects** Symptom control Being physically disabled Phases of the progressive disease	**Needs concerning autonomy** Self-determination and choice Dependency Limitations of activities in everyday life
**Needs concerning emotions** Emotional distress Anxious feelings Feelings of loss Gratitude	**Needs concerning coping with the disease** Acceptance of the palliative condition Prognostic awareness and hope Changes in goals of care and treatment decisions Adjustment to stages of the illness trajectory
**Needs concerning existential and spiritual aspects** Spirituality Life review Inner peace Desire to die	**Needs concerning delivery/quality of care** Issues of accessing and receiving care in the past Perceptions of palliative care quality Next steps in palliative care Security
**Needs concerning social aspects** Social relationships Family caregiver burden Needs emerging from the individual biography	**Needs concerning preparations for death** Bureaucracy including setting up powers of attorney Planning for funeral arrangements

## Data Availability

The data that support the findings of this study are available on request from the corresponding author (CG). The data are not publicly available because they contain information that could compromise the privacy of research participants.

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
