# Peer review of "The Impact of the SARS-CoV-2 Pandemic on the Needs of Non-Infected Patients and Their Families in Palliative Care—Interviews with Those Concerned"

_jcm, 2022, doi:10.3390/jcm11133863_

Round 1

Reviewer 1 Report

The pandemic has impacted every aspect of the healthcare system, including delivery of palliative care, which is important to investigate.  A couple of minor comments to strengthen the manuscript

1) Based on allowing providers to decide a patient was too overwhelmed with care needs, it would be good to have a sense of how many patients were screened due to such "gatekeeping" and could be discussed in the limitations how such gatekeeping might have affected the results.

2) Figure 1 is basically the same as Figure 2 , but with a little less information - you can just use Figure 1

3) It would be nice if the results had a little less text and if some of the quotes were shorter.  It would help highlight the key findings better and really, only 1 quote is needed in the text per section - you can include others in a table.
